# POSITIVE AND UNLABELED LEARNING INCORPORATING ADDITIONAL POSTERIOR PROBABILITIES

## ABSTRACT

Learning from Positive and Unlabeled (PU) data presents unique challenges in scenarios where negative examples are absent. Many state-of-the-art PU methods are prior-based which assumes that the class probability within the unlabeled data corresponds to the class prior probability. However, this framework often falls short when attempting to accurately represent the complexities of real-world applications, such as industrial anomaly detection, where variations in data distribution within the combined training set are prevalent. In this paper, we introduce a generalized PU framework that models uncertainty via subset-specific posterior probabilities, proposing a posterior-based method (postPU) with theoretically and empirically validated consistency. Further, we establish that sample weighting is fundamental to PU robustness and derive a class-balanced weighting principle to minimize sensitivity to label inaccuracies. Experiments show the effectiveness and robustness of postPU and its capacity to leverage auxiliary uncertain annotations.

## 1 INTRODUCTION

Positive and Unlabeled (PU) learning trains a binary classifier using only positive and unlabeled data, without explicit negative examples Liu et al. (2003). This approach is critical in domains where negative labels are difficult to obtain Bekker & Davis (2020); Jaskie & Spanias (2019); Gong et al. (2025). For instance, in bioinformatics, samples that exhibit certain properties can be labeled as positive, while those that lack such properties cannot be confidently labeled as negative Li et al. (2021). Similarly, in personalized advertising, visited links are considered as positive samples of user interest, whereas non-visited links are not necessarily uninterested Bekker & Davis (2020). For anomaly detection scenarios in industries, the operating status of machines (e.g. wind turbines) is difficult to be labeled as normal or anomalous, especially in the deterioration stage Qian et al. (2023); Zheng & Zhao (2022).

The unbiased PU risk estimator (uPU) Plessis et al. (2014) provides a foundation for PU learning under the case-control and Selected Completely at Random (SCAR) assumptions Menon et al. (2015); Elkan & Noto (2008). Within this framework, misclassification risk is estimated indirectly, and a classifier is trained via empirical risk minimization (ERM). This approach treats unlabeled data as i.i.d. samples from the marginal distribution and requires the class prior to be known or estimable Elkan & Noto (2008); Plessis & Sugiyama (2014); Plessis et al. (2017); Nakajima & Sugiyama (2023). The PU risk estimator, central to this framework, derives the risk for negative examples by combining risks from positive and unlabeled data.

However, prior-based PU learning methods face significant challenges when abstracting real-world problems into the standard PU framework, which requires defining Positive/Unlabeled subsets and estimating an overall class prior. First, inaccurate estimation of the class prior within the large unlabeled set can substantially degrade classification performance. Moreover, the assumption that the unlabeled set conforms to the true marginal distribution is often overly idealistic. Consequently, effectively addressing the dual challenges of inaccurate class prior estimation and limited representation in training labels is crucial for the successful real-world application of PU learning methods.

To this end, we introduce an extension of the prior-based PU learning problem as posterior-based PU learning. This extended framework adheres to the general ERM framework, encompassing the traditional prior-based problem as a special case. It allows for a more nuanced representation of

the training data by representing label uncertainty by multiple class posterior probabilities, taking into account the corresponding evidence. Moreover, we propose a novel posterior-based PU risk estimator (postPU) for this extended framework. Our main contributions can be summarized as follows:

**Generalized PU framework**: We introduce a generalized PU learning formulation that enables nuanced modeling of uncertainty in the unlabeled set through subset-specified posterior probabilities. A corresponding method , denoted *postPU*, is proposed, with its convergence and consistency established through theoretical analysis and empirical validation.

**Insights on sample weighting**: We demonstrate that sample weights affect the consistency and robustness of PU learning. Consequently, we propose a class-balanced weighting principle to maximize robustness against class probability estimation errors. This principle is supported by both theoretical guarantees and experimental validation.

**Experiments and applications**: We provide quantitative evaluations demonstrating the effectiveness and robustness of *postPU*. Additionally, we show that its capacity to incorporate additional posteriors facilitates the utilization of auxiliary uncertain annotations, thereby enhancing classification performance. This capability finds applications in industrial anomaly detection scenarios with uncertain and imperfect annotations.

## 2 PRELIMINARIES: PRIOR-BASED PU LEARNING

In PU learning, training samples $\boldsymbol{x} \in \mathcal{R}^d$ are partitioned into a positive set $\mathcal{X}_p$ and an unlabeled set $\mathcal{X}_U$. The class prior probability $\pi = P(Y = Y^+)$ is either known a priori or estimated from the dataElkan & Noto (2008); Plessis & Sugiyama (2014); Plessis et al. (2017); Nakajima & Sugiyama (2023). Central to prior-based PU learning methods is the Select Completely At Random (SCAR) assumptionElkan & Noto (2008). This assumption posits that data samples from unlabeled set $\mathcal{X}_u = \{\boldsymbol{x}_i^u\}_{i=1}^{n_u}$ are drawn identically and independently from the marginal distribution $p(\boldsymbol{x})$, without regard of their features. In binary classification, the learning objective is: $\mathcal{R} = \pi \mathcal{R}_+^+ + (1 - \pi)\mathcal{R}_-^-$ where, the risks associated with the negative and positive classes are defined by: $\mathcal{R}_+^+ = \mathbb{E}_{\boldsymbol{x} \sim p^+}[l(g(\boldsymbol{x}), y^+)]$ and $\mathcal{R}_-^- = \mathbb{E}_{\boldsymbol{x} \sim p^-}[l(g(\boldsymbol{x}), y^-)]$. A summary of general notations with definitions is provided in the supplementary material A.

A major challenge is estimating negative risk in the absence of negative labels. Under the SCAR assumption, the latent positive samples in the unlabeled set are distributed according to the same marginal distribution as the labeled positives. This assumption enables leveraging the known class prior probability $\pi$ to deduce the negative risks from absolute risks by $(1 - \pi)\mathcal{R}_-^-(g) = \mathcal{R}_u^-(g) - \pi \mathcal{R}_+^-(g)$. The *absolute positive risk* $\mathcal{R}_u^+ \in [0, 1] := \mathbb{E}_{\boldsymbol{x} \sim p}[l(g(\boldsymbol{x}), y^+)]$ and *absolute negative risk* $\mathcal{R}_u^- \in [0, 1] := \mathbb{E}_{\boldsymbol{x} \sim p}[l(g(\boldsymbol{x}), y^-)]$ are defined based on misclassfication risks assuming uniform positive/negative as true labels. These absolute risks can be computed without training labels. By substituting the negative risk in the objective of supervised positive-negative(PN) risk estimation from learning objective with the above expressions, the unbiased PU estimator is obtained:

$$\mathcal{R}_{uPU}(g) = \underbrace{\pi \mathcal{R}_+^+(g)}_{\text{positive risk}} + \underbrace{\mathcal{R}_u^-(g) - \pi \mathcal{R}_+^-(g)}_{\text{negative risk}}. \tag{1}$$

This unbiased PU risk estimator was first proposed by Plessis et al. (2014). Then Kiryo et al. (2017) integrated a non-negative constraint to negative risk to prevent over-fitting when using deep models and proposed nnPU. Currently, most prior-based PU learning methods are based on the nnPU risk estimator and introduce various augmentations to expand its scope of application or improve its performance Kato et al. (2018); Hsieh et al. (2018); Watanabe & Matsui (2023). For instance, ImbPU Su et al. (2021) that targets specifically at imbalanced data, PULDA Jiang et al. (2023) that introduces confidence penalization terms to enhance discriminability, SSLPU Wang et al. (2022) leverages semi-supervised learning framework to tackle the negative sample misclassification problem.

## 3 POSTERIOR-BASED PU LEARNING FORMULATION

In the context of prior-based PU learning, the absence of negative class labels is acknowledged; it is compensated by the assumption that the class probability within the unlabeled set is known, i.e., $P(y = y^+|x \in \mathcal{X}_U) = \pi_U^+ = P(y = y^+)$. However, we argue that the latter part that assumes the class posterior probability corresponds to the class prior probability $\pi_U^+ = P(y = y^+)$ is not necessarily accurate or a mandatory condition for PU learning. The class posterior probability $P(y = y^+|x \in \mathcal{X}_U)$ represents the data distribution of training samples, influenced by the accessibility of samples in data collection. In contrast, the class prior probability $P(y = y^+)$ represents the data distribution in test scenarios.

The unbiased PU risk estimator remains valid even when $\pi_U^+ \neq P(y = y^+)$, though it may introduce a bounded error (we analyze this error via effective class ratio specified by Eq. (4) ). Thus, we abandon this assumption and redefine the fuzzy class label in the unlabeled set based directly on the class posterior probability $P(y = y^+|x \in \mathcal{X}_U)$. Moreover, the assumption in prior-based PU learning that the unlabeled set adheres to a uniform distribution may be overly simplistic Bekker et al. (2020). So we extend the problem setting to accommodate a list of class posterior probabilities that divide samples into $m \geq 2$ subsets, aiming to address the diversity in data accessibility inherent in semi-supervised learning and maximize utilization of label information.

Assume the training set provides class labels as a list of posteriors that describe the class probability under several pieces of evidence. The training set is segmented into subsets based on this evidence, denoted as $\mathcal{X}_i$ for the $i$th subset, with class posterior probability $P(y = y^+|\boldsymbol{x} \in \mathcal{X}_i) = \pi_i$ (estimated empirically or statistically). This framework provides a general form for multi-subset learning scenarios, in which the traditional PU learning problem is subsumed as a special case defined by $m = 2$ subsets, with a positive subset having a class posterior of $\pi_1 = 1$ and an unlabeled subset with class posterior $\pi_2 = \pi$. The inherent robustness of postPU thereby facilitates training using coarsely approximated $\pi_i$ values. For instance, when partitioning data into $m = 3$ subsets—namely likely negative, ambiguous, and likely positive categories—these are characterized by progressively increasing probabilities satisfying $\pi_1 < \pi_2 < \pi_3$.

Following the idea of unbiased PU risk estimator defined by Eq.(1), we estimate PU risks in posterior-based PU learning setup by weighted sum of absolute risks in each subset:

$$\mathcal{R} = \underbrace{\sum_{i=1}^{m} \lambda_i^+ \mathcal{R}_i^+}_{\text{positive risk}} + \underbrace{\sum_{i=1}^{m} \lambda_i^- \mathcal{R}_i^-}_{\text{negative risk}} . \tag{2}$$

The determination of *sample weights* $\lambda_i^+$ and $\lambda_i^-$ is key to accuracy and consistency of risk estimation. We substantiate this argument both theoretically and empirically in the following subsections.

### 3.1 CLASS-BALANCED WEIGHTING PRINCIPLE

We begin by analyzing how class probability estimation errors can lead to inaccuracies in PU risk estimation and consequently impair the overall performance of the learning methods. In both prior-based risk estimator Eq. (1) and posterior-based risk estimator (2) the absolute risks $\mathcal{R}_i^+, \mathcal{R}_i^-$ are expressed as linear combinations of true risks. We then express them by a weighted combination of true positive risk and true negative risk:

$$\mathcal{R}_{postPU} = \mathcal{R}_{postPU}^+ + \mathcal{R}_{postPU}^- = k^+ \mathcal{R}_+^+ + k^- \mathcal{R}_-^- + C \tag{3}$$

We define effective class weight $\bar{r} = k^+/(k^+ + k^-)$ as the proportion of true positive risk in the whole risk. Then we have the following theorem, demonstrating how it dominate the optimal classifier:

**Theorem 1.** *Let $g_1^*$ and $g_2^*$ be two classifiers that minimize two risks respectively on the same data distribution $p$:*

$$g_c^* = \arg\min_{g \in \mathcal{G}} \mathcal{R}_c(g, p), \mathcal{R}_c(g, p) = k_1^+ \mathcal{R}_+^+(g, p) + k_1^- \mathcal{R}_-^-(g, p) + c_1(k_1^+ > 0, k_1^- > 0), c \in \{1, 2\}.$$

*If $k_1^+/k_1^- = k_2^+/k_2^-$ we have $g_1^* = g_2^*$.*

Proof and derivation of theorem 1 is provided in supplementary material C.1. Since $\frac{k^+}{k^-} = \frac{\overline{r}}{1-\overline{r}}$, theorem 1 shows that the optimal classifier $g^*$ varies with the linear coefficients of the risk estimator $\mathcal{R}$ in a way that it depends solely on the effective class weight $\overline{r}$. When overlooking the magnitude of risks that pertain exclusively to the learning rate, risk estimators sharing the same $\overline{r}$ value are deemed equivalent.

**Robustness with class probability estimation error:**

Suppose we have weights $\lambda_i^+, \lambda_i^-$ in Eq.(3) determined by estimated class probabilities $\hat{\pi}_i$ while the true class probabilities are $\pi_i$. Then the true effective class weight $\hat{\overline{r}}$ is given as:

$$\hat{\overline{r}}(\boldsymbol{\pi}) = (\sum_{i=1}^{m} \lambda_i^+ \pi_i - \sum_{i=1}^{m} \lambda_i^- \pi_i)/[\sum_{i=1}^{m} \lambda_i^+ \pi_i - \sum_{i=1}^{m} \lambda_i^- \pi_i - \sum_{i=1}^{m} \lambda_i^+ (1-\pi_i) + \sum_{i=1}^{m} \lambda_i^- (1-\pi_i)].$$

(4)

The impact of class probability estimation error on effective class weight can be quantitatively estimated by $e_{\overline{r}}(\hat{\boldsymbol{\pi}}, \boldsymbol{\pi}) = \hat{\overline{r}}(\boldsymbol{\pi}) - \hat{\overline{r}}(\hat{\boldsymbol{\pi}})$. We have the following results (proof is provided in the supplementary material C.2):

**Theorem 2.** *For true effective class probability $\hat{\overline{r}}(\boldsymbol{\pi})$ defined by Eq. (4), $\forall \boldsymbol{\pi} \in D, \hat{\overline{r}}(\boldsymbol{\pi}) = \hat{\overline{r}}(\hat{\boldsymbol{\pi}})$, if and only if $\sum_{i=1}^{m} \lambda_i^+ - \lambda_i^- = 0$.*

**Theorem 3.** *Given a posterior-based risk estimator $\mathcal{R}_{postPU}(g, p, \boldsymbol{\lambda}, \boldsymbol{\pi})$ defined by Eq.(3). Define $g^*(\boldsymbol{x})$ the optimal classifier satisfying $g^*(\boldsymbol{x}) = \arg\min_{g \in \mathcal{G}} \mathcal{R}_{postPU}(g, p, \boldsymbol{\lambda}, \hat{\boldsymbol{\pi}})$ using the subsets' estimated class posteriors $\hat{\boldsymbol{\pi}}$, and $g_o^*(\boldsymbol{x})$ the oracle optimal classifier satisfying $g_o^*(\boldsymbol{x}) = \arg\min_{g \in \mathcal{G}} \mathcal{R}_{postPU}(g, p, \boldsymbol{\lambda}, \boldsymbol{\pi})$ using the subsets' true class posteriors $\boldsymbol{\pi}$. If the coefficients satisfy $\sum_{i=1}^{m} \lambda_i^+ - \lambda_i^- = 0, \forall \boldsymbol{\pi} \in D, g^* = g_o^*$*

*Proof.* Combining theorem 1 and theorem 2 proofs theorem 3. $\qquad\square$

Theorem 3 demonstrates that by satisfying the class-balanced weighting principle, the risk estimator can be resistant to inaccuracies of class probabilities. Specifically, it posits that a postPU risk estimator, employed to train classifiers with class probabilities that contain significant estimation error, is expected to perform equivalently to an oracle risk estimator that utilizes the true class probabilities. By regulating the magnitude of $\lambda$, we transform theorem 3 into the following class-balanced weighting principle:

$$\begin{cases} \sum_{i=1}^{m} \lambda_i^+ \pi_i = \frac{1}{2}, \\ \sum_{i=1}^{m} \lambda_i^- (1-\pi_i) = \frac{1}{2}. \end{cases}$$

(5)

**Illustration:** We illustrate the influence of class probability estimation error on a synthetic PU problem (ground truth class labels and ideal separation line illustrated in Fig. 1(a) ). We assess the performance of nnPU and postPU in this PU learning problem by training a Multi-Layer Perception (MLP) classifier, while considering a class probability estimation error $\delta = \hat{\pi} - \pi$.

This prior-based PU learning problem is a special case of posterior-based PU learning problem where $m = 2, \pi_0 = \pi, \pi_1 = 1$. The equivalent choices of $\lambda_i^+, \lambda_i^-$ in nnPU are listed in Table 1, in comparison with those of postPU. We express the true effective class weight of both estimators in terms of the true class prior $\pi$ and the class probability estimation error $\delta$. The effective prior of nnPU varies with the estimated class probability, while that of postPU remains invariant at 1/2 (illustrated by Fig. 1(c)).

Fig. 1(d) compares the experimental results of nnPU and postPU with class probability estimation error on the synthetic data mentioned above. F1-scores of postPU (solid lines), remain relatively stable as the estimated probability $\hat{\pi}$ varies from 0.1 to 0.9 . The results demonstrate the robustness of postPU in the presence of class probability estimation error.

Table 1: Robustness of PU risk estimators with class probability estimation error.

| | $\pi_i$ | $\hat{\pi}_i$ | $\lambda_i^+$ | $\lambda_i^-$ | $k^+$ | $k^-$ | $\pi + \delta$ | $\overline{r}$ |
|---|---|---|---|---|---|---|---|---|
| nnPU(P) | $\pi$ | $\pi + \delta$ | $0$ | $1$ | $\pi + 2\delta$ | $1 - \pi - \delta$ | $(\pi/2, 1)$ | $\dfrac{\pi + 2\delta}{1 + \delta}$ |
| nnPU(U) | $1$ | $1$ | $\pi + \delta$ | $1 - \pi - \delta$ | | | | |
| postPU(P) | $\pi$ | $\pi + \delta$ | $0$ | $1/[2(1 - \pi - \delta)]$ | $1/2$ | $1/2$ | $(0, 1)$ | $1/2$ |
| postPU(U) | $1$ | $1$ | $1/2$ | $(\pi + \delta)/[2(1 - \pi - \delta)]$ | | | | |

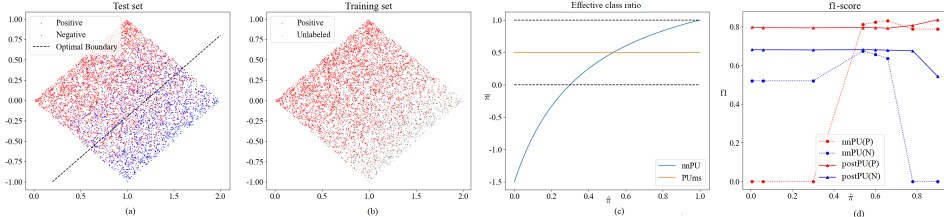

Figure 1: Illustrative experiment comparing robustness in terms of class probability estimation error between nnPU and postPU: (a) test data with ground truth label; (b) PU training set; (c) effective class weight that varies with estimated class probabilities; (d) performance in terms of f1 score with corresponding estimated class probabilities.

### 3.2 INCORPORATING AUXILIARY UNCERTAIN ANNOTATIONS

In this subsection, we first investigate the influence of sample weights $\lambda$ on the generalization error of the posterior-based risk estimator. Subsequently, we provide an experimental comparison between the traditional $m = 2$ setup and the $m = 3$ configuration, which incorporates additional evidence as annotation information.

**Bound for variance of generalization error:** By assuming that all training samples $\boldsymbol{x}_{i,j}$ are independent, we study the convergence rate of empirical risk estimators using its variance:

$$\mathrm{Var}(\hat{\mathcal{R}}_{postPU}) = \sum_{i=1}^{m} \frac{(\lambda_i^+)^2}{n_i} \sum_{j=1}^{n_i} \mathrm{Var}[l(g(\boldsymbol{x}_{i,j}), y^+)] + \sum_{i=1}^{m} \frac{(\lambda_i^-)^2}{n_i} \sum_{j=1}^{n_i} \mathrm{Var}[l(g(\boldsymbol{x}_{i,j}), y^-)]. \quad (6)$$

Since loss function $l(g, y)$ and its variance are bounded in $[0, 1]$, an upper bound of Eq.(6) is derived as:

$$\mathrm{Var}(\hat{\mathcal{R}}_{postPU}) \leq \sum_{i=1}^{m} \frac{(\lambda_i^+)^2}{n_i} + \frac{(\lambda_i^-)^2}{n_i}. \quad (7)$$

**Effective training set size:** Eq. (7) indicates that the variance of the postPU empirical risk estimator $\mathrm{Var}(\hat{\mathcal{R}}_{postPU}) \to 0$ in $\mathcal{O}(1/n)$. The selection of sample weights $\lambda_i^+, \lambda_i^-$ affects the absolute value of the above variance, especially with highly imbalanced training data. We define *effective training set size* as the reciprocal of the upper bound of variance of the empirical risk by $N(\mathcal{R}, X, Y) = 1/\sup_{g \in \mathcal{G}} \mathrm{Var}[\mathcal{R}(g, X, Y)]$. It indicates the required training set size for a binary classifier on fully supervised data equivalent to the variance of PU risk estimator $\mathcal{R}$ on data $(X, Y)$. Since expectations of PU risks are upper bounded by 1, $N$ quantitatively evaluates the data utilization efficiency of a PU risk estimator on a given training set.

The analytical results of effective training set size for nnPU and postPU($m = 2, 3$) are detailed in supplementary material D. They indicate that in traditional two-subset ($m = 2$) PU learning scenarios, the above variance is dominated by the size of the minority set. Specifically, as the size of majority set $n_i$ approaches infinity, the effective training set size $N$ converges to $Cn_i$. Therefore, we propose to minimize this variance by minimizing an objective function:

$$\sum_{i=1}^{m} (\lambda_i^+)^2/n_i + (\lambda_i^-)^2/n_i. \quad (8)$$

Table 2: Effective training set sizes in PU learning settings.

|        | method | $\pi_1$ | $n_1$ | $\pi_2$ | $n_2$ | $\pi_3$ | $n_3$ | $N$ | $F1$ |
|--------|--------|---------|-------|---------|--------|---------|---------|---------|--------|
| case A | nnPU   | 100%    | 500   | -       | -      | 10%     | 100,000 | 606.1   | 0.0%   |
| case B | postPU | 100%    | 500   | -       | -      | 10%     | 100,000 | 1963.6  | 55.59% |
| case C | postPU | 100%    | 500   | 50%     | 10,000 | 5.56%   | 90,000  | 10187.2 | 89.48% |

**Illustration:** Numerical experiments are conducted on a synthetic moon-shaped dataset with 0.18 noise and 24% mislabeled rate to further substantiate the above findings. Table 2 lists numerical experiments on a synthetic two-moon dataset, with Fig. 2 visualizing predictions. Case C adds an ambiguously labeled set ($\pi_2 = 0.5 > \pi$), obtained by labeling likely positives ($\pi_2 = 0.5$) based on uncertain evidence within $\mathcal{X}_U$ without introducing additional samples. Notably, the resulting effective training set size $N$ in case A is comparable to the size of the minority set $n_1 = 500$, demonstrating a minor impact of the extensive unlabeled data on the classification stability. The contrast between case B and case C demonstrates that postPU promotes classification stability via incorporating additional evidence and posterior probabilities. This increase of $N$ is notable despite the imprecision in the labeling evidence ($\pi_2 = 50\%$).

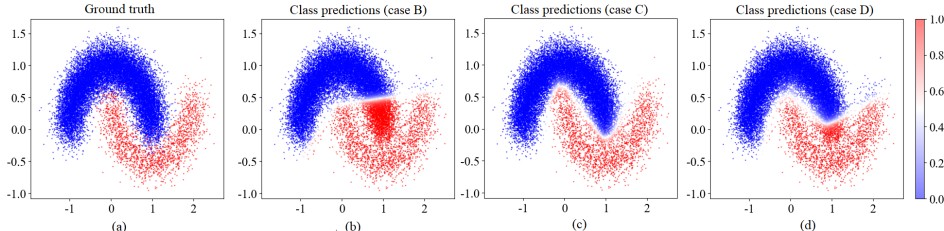

Figure 2: Illustrative experiment of postPU utilizing $m = 3$ subsets with different levels of label precision. (a) Ground truth; (b) case B $m = 2$; (c) case C with an additional ambiguously labeled set; (d) case D that ablates the optimal selection of weights in case C.

We examine the benefits of Eq. (8) via an ablation study (case D). Here, weights $\lambda_i^+$ and $\lambda_i^-$ are optimized assuming uniform training set sizes, rather than based on actual sizes. This results in a significantly smaller effective training set ($N = 2520.9$) and correspondingly lower classification performance (F1-score: 78.18%). The contrast with case C demonstrates that weights finely attuned to sample numbers yield substantial benefits, particularly on highly imbalanced datasets where appropriate weighting mitigates imbalance effects.

## 3.3 ALGORITHM

In the previous analysis, we derive two constraints Eq. (5) and (8) for the generalized risk estimator Eq. (2). Additionally, similar to prior-based risk estimators, the estimation of positive and negative risk should be aligned with the true misclassification risk of positive and negative class. This leads to the primary constraint:

$$\begin{cases} \sum_{i=1}^m \lambda_i^+ (1 - \pi_i) = 0, \\ \sum_{i=1}^m \lambda_i^- \pi_i = 0. \end{cases} \quad (9)$$

Finally, we introduce postPU, a novel PU risk estimator. Integrating the constraints in Eq. (9), (5) and (8), sample weights $\lambda_i^+$ and $\lambda_i^-$ can be solved using known posterior probabilities $\pi_i$ and subset sizes $n_i$. Substituting these weights into Eq. (2) yields the postPU estimator. Analogous to prior-based methods (e.g., nnPUKiryo et al. (2017)), we use postPU as the loss function within the ERM framework, solving posterior-based PU learning. This native ERM approach readily incorporates modern techniques like meta-learning and self-learning; we leave such integration for future studies.

# 4 IMPLEMENTATION

A practical algorithm for solving sample weights as a pseudo-code of the overall ERM framework is demonstrated below. Details are specified in the supplementary material B.

---

**Algorithm 1** postPU

---

**Require:** Training samples $\boldsymbol{x}_{i,j}$ in subsets $\mathcal{X}_i$, each with total sample number of $n_i$ and a distinct class posterior of $\pi_i$; classification model space $\mathcal{G}$; symmetric surrogate binary classification loss function $l$; hyperparameter $\beta$.
**Ensure:** Optimized classifier $g \in \mathcal{G}$.
1: Solve $\lambda_i^+$ and $\lambda_i^-$ according to Eq.(5), (9) and (8)
2: Randomly initialize $g$
3: **for** each epoch **do**
4:   **for** each batch $\mathcal{B}_t$ **do**
5:     **for all** subset indices $i$ in $1, 2, \dots, m$ **do**
6:       $\hat{\mathcal{R}}_{t,i}^+ \leftarrow \underset{\boldsymbol{x} \in \mathcal{B}_t \wedge \boldsymbol{x} \in \mathcal{X}_i}{Average} \, l[g(\boldsymbol{x}), y^+] \, ; \hat{\mathcal{R}}_{t,i}^- \leftarrow 1 - \hat{\mathcal{R}}_{t,i}^+$  {absolute risks}
7:     **end for**
8:     $\hat{\mathcal{R}}_{postPU,t}^+ \leftarrow \sum_{i=1}^m \lambda_i^+ \hat{\mathcal{R}}_{t,i}^+ \, ; \hat{\mathcal{R}}_{postPU,t}^- \leftarrow \sum_{i=1}^m \lambda_i^- \hat{\mathcal{R}}_{t,i}^-$  {positive, negative risks}
9:     **if** $\hat{\mathcal{R}}_{postPU,t}^+ < 0$ **then**
10:       $\hat{\mathcal{R}}_{postPU,t} \leftarrow -\beta \hat{\mathcal{R}}_{postPU,t}^+$  {Non-negative}
11:     **else if** $\hat{\mathcal{R}}_{postPU,t}^- < 0$ **then**
12:       $\hat{\mathcal{R}}_{postPU,t} \leftarrow -\beta \hat{\mathcal{R}}_{postPU,t}^-$  {Non-negative}
13:     **else**
14:       $\hat{\mathcal{R}}_{postPU,t} \leftarrow \hat{\mathcal{R}}_{postPU,t}^+ + \hat{\mathcal{R}}_{postPU,t}^-$  {Learning objective}
15:     **end if**
16:     Employ gradient descent according to objective $\hat{\mathcal{R}}_{postPU,t}$ to update $g$
17:   **end for**
18: **end for**

---

# 5 THEORETICAL ANALYSIS

We analyze the generalization error of postPU risk estimator in posterior-based PU learning by giving an estimation error bound:

**Theorem 4.** *Assume that the loss $l[g(\boldsymbol{x}), y]$ is symmetric and $\phi$-Lipschitz continuous with respect to its first argument. For any $\delta > 0$ with probability at least $1 - \delta$, we have:*

$$\mathcal{R}(\hat{g}) - \mathcal{R}(g^*) \leq \sum_{i=1}^m 4(\lambda_i^+ + \lambda_i^-)\phi_g \mathfrak{R}_{n_i, V_i}(\mathcal{G}) + \sum_{i=1}^m (\lambda_i^+ + \lambda_i^-)\sqrt{\frac{2\ln(4/\delta)}{n_i}} \qquad (10)$$

This generalization error bound implies $\mathcal{R}(\hat{g}) \to \mathcal{R}(g^*)$ in $\mathcal{O}(m/\sqrt{\min n_i})$ if $\mathfrak{R}$ is upper bounded for $\mathcal{G}$. This bound for postPU is no worse than prior-based PU estimators Plessis et al. (2014); Kiryo et al. (2017); Su et al. (2021). An asymptotic analysis of the computational cost and the proof of this theorem is provided in the supplementary material E.

# 6 EXPERIMENTS

## 6.1 COMPARATIVE EXPERIMENTS

We experimentally compare the performance of postPU with state-of-the-art PU learning methodsKiryo et al. (2017); Su et al. (2021); Long et al. (2024); Zhu et al. (2023); Jiang et al. (2023) on a PU problem with additional ambiguous labels on two benchmark computer vision datasets CIFAR-10 Krizhevsky (2009) and F-MNIST Xiao et al. (2017) (specified in the supplementary material F). This experimental setup simulates a semi-supervised learning scenario in which a small, accurately

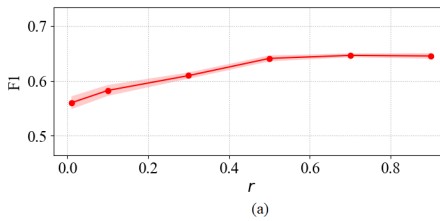 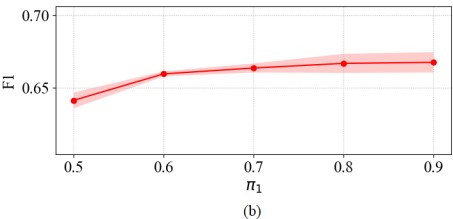

Figure 3: Sensitivity analysis of postPU. (a) Performance of postPU that varies with label ratio $r$ ; (b) performance of postPU that varies with posterior $\pi_1$.

labeled positive dataset $\mathcal{X}_P$ is available, alongside a larger unlabeled dataset $\mathcal{X}_U$. Moreover, we introduce an automatic, albeit imperfect, labeling method that identifies samples meeting a specific criterion $V$. This evidence divides $\mathcal{X}_U$ into two subsets: $\{\boldsymbol{x}|V \wedge \boldsymbol{x} \in \mathcal{X}_U\}$ and $\{\boldsymbol{x}|\neg V \wedge \boldsymbol{x} \in \mathcal{X}_U\}$.

The classification F1-score of postPU and other comparison methods are reported in Table 3. When compared to other native ERM methods (nnPU and ImbPU) postPU achieves significantly higher performance. Additionally, postPU outperforms the remaining state-of-the-art methods with meta-learning enhancements. This indicates that the postPU method effectively utilizes the auxiliary information provided by $V$ to improve classification performance. Notably, postPU effectively leverages this uncertain annotation evidence $V$ which is only marginally better than random selection ($\pi_1 = 50\%$ over $\pi_p = 40\%$).

Table 3: Results of F1-score (mean±std) of comparative experiment on benchmark datasets

| DataSet | Native ERM | | | Beyond ERM | | |
|---|---|---|---|---|---|---|
| | postPU | ImbPU | nnPU | LaGAM | robustPU | DistPU |
| CIFAR-10 | **64.4 ± 0.5** | 55.2 ± 1.9 | 34.5 ± 7.5 | 57.0 ± 2.3 | 55.4 ± 0.2 | 49.1 ± 2.6 |
| F-MNIST | **94.7 ± 0.1** | 94.4 ± 0.1 | 93.5 ± 0.1 | 79.7 ± 6.5 | 88.7 ± 0.1 | 91.7 ± 0.3 |

## 6.2 ROBUSTNESS AGAINST INACCURATE GUESSES OF POSTERIORS

Building upon the theoretical guarantees provided by Theorem 3 regarding robustness to class probability estimation errors, the proposed postPU framework enables effective training with only coarsely approximated posterior probabilities. To empirically validate the theoretical claims of Theorem 3, we conduct the following experiment. Suppose we have access to class label evidence with associated uncertainty, which leads us to partition the training set into three subsets: likely negatives, ambiguous samples, and likely positives. The posterior probabilities of positive samples in these subsets are denoted as $\pi_1 < \pi_2 < \pi_3$ respectively. Table 4 demonstrates the F1-score performance of postPU on CIFAR-10 and F-MNIST using using roughly guessed posterior probability estimates $\hat{\pi}_1, \hat{\pi}_2, \hat{\pi}_3$. For reference, the last column provides baseline results obtained with accurate posterior probabilities.

Table 4: Results of F1-score (mean±std) of postPU using roughly guessed posterior probabilities.

| $(\hat{\pi}_1, \hat{\pi}_2, \hat{\pi}_3)$ | $(.3, .6, .9)$ | $(.25, .5, .75)$ | $(0, .5, 1)$ | $(0, .25, .5)$ | $(.5, .75, 1)$ | $(.33, .5, 1)$ |
|---|---|---|---|---|---|---|
| CIFAR-10 | 64.6 ± 0.5 | 64.7 ± 0.4 | 64.6 ± 0.5 | 64.7 ± 0.3 | 64.4 ± 0.5 | 64.4 ± 0.5 |
| F-MNIST | 95.1 ± 0.4 | 95.0 ± 0.7 | 94.9 ± 0.3 | 95.3 ± 0.3 | 95.0 ± 0.4 | 95.7 ± 0.1 |

The results in Table 4 demonstrate that the performance of postPU remains robust across various guessed $\hat{\pi}_i$. This indicates that postPU does not rely on accurate estimation of the actual posterior probabilities $\pi_i$. Instead, it successfully leverages the fuzzy labeling information provided by the relative ordering of posterior probabilities ($\pi_1 < \pi_2 < \pi_3$). The distinctiveness of the posterior probability estimates correlates with the quality of the fuzzy labeling information, with greater distinctions leading to higher information value.

Additional experiments are to evaluate the impact of varying $r$ (Fig. 3(a)) and $\pi_1$ (Fig. 3(b)) on the performance of postPU. The label ratio $r$ indicates the proportion of true positives that meet

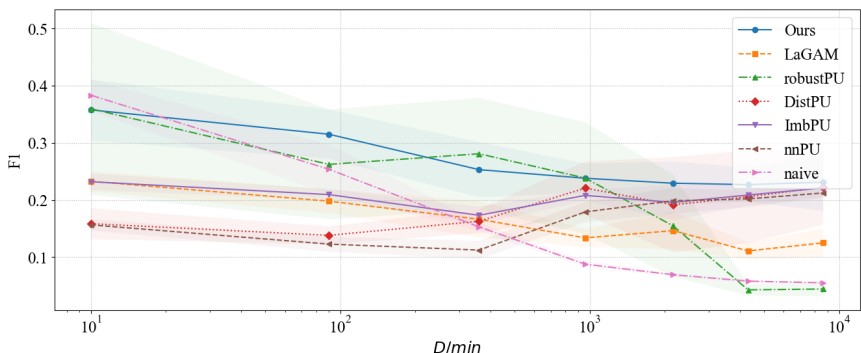

Figure 4: Quantitative analysis of robustness of compared methods against annotation uncertainty (maximum discrepancy $D$).

the evidence $V$. By varying $r$ from 0.01 to 0.9, an increasing trend in classification performance is observed. This result is anticipated, as a larger $r$ implies a higher proportion of samples fuzzily labeled by $V$. The posterior probability $\pi_2 = P(y = y^+|V \wedge \boldsymbol{x} \in \mathcal{X}_U)$ reflects the precision of $V$ as a pseudo-positive label.

### 6.3 APPLICATION ON INDUSTRIAL ANOMALY DETECTION UNDER INACCURATE LABELS

We extend our study with an application experiment on wind turbine anomaly detection, utilizing a publicly available dataset Leahy et al. (2016). A critical challenge in wind turbine fault detection arises from the inherent inconsistency between actual faults and failure records. Systematic factors such as maintenance cycles and delayed human responses typically induce recording discrepancies spanning several days. Denoting $\tau_i^*$ as the true failure occurrence time and $\tau_i$ as the recorded time, the discrepancy $\Delta\tau_i = \tau_i - \tau_i^*$ exhibits quantifiable non-random patterns through prior knowledge. We apply postPU method by designating training samples within the discrepancy window $(\hat{\tau}_i - D, \hat{\tau}_i)$ as likely faults. These probabilistically labeled instances, together with definitive normal ($\pi = 0$) and faulty ($\pi = 1$) samples, establish a tripartite training structure ($m = 3$) with posterior probabilities $0, 1, \pi_l$. Specific experiment setting is demonstrated in the supplementary material F. Fig. 4 quantifies robustness against label uncertainty (maximum discrepancy $D$). The naive baseline exhibits progressive performance degradation with increasing $D$, highlighting the detrimental impact of uncertain annotations. In contrast, PU methods explicitly account for false positives during training, substantially reducing false alarms. The proposed method demonstrates superior robustness to label discrepancies among the comparison methods.

## 7 CONCLUSIONS

This work advances Positive-Unlabeled (PU) learning through a generalized framework that incorporates uncertain annotations via subset-specific class posterior probabilities. Our theoretical analysis reveals that the class-balanced weighting principle within the proposed *postPU* method guarantees its robustness against class probability estimation error. Theoretical analysis and empirical evaluations demonstrate *postPU*'s superior effectiveness when leveraging auxiliary posterior information. Experiments also underscores the framework's practical value for industrial applications, particularly in anomaly detection under imperfect supervision characterized by label ambiguity.

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

SUPPLEMENTARY MATERIALS

This section presents additional proof and experiment setting in this paper.

## A    TABLE OF NOTATIONS

Summary of general notations with definitions is listed in Table 7.

## B    IMPLEMENTATION

In the implementation of postPU for training deep learning models, a non-negative constraintKiryo et al. (2017) should be employed to prevent over-fitting:

$$\mathcal{R}_{postPU} = \max(0, \sum_{i=1}^{m} \lambda_i^+ \mathcal{R}_i^+) + \max(0, \sum_{i=1}^{m} \lambda_i^- \mathcal{R}_i^-). \tag{11}$$

In practice, the absolute risks are substituted with their empirical counterparts, calculated as follows:

$$\hat{\mathcal{R}}_i^+ = \frac{1}{n_i} \sum_{j=1}^{n_i} l[g(\boldsymbol{x}_{i,j}), y^+],$$

$$\hat{\mathcal{R}}_i^- = \frac{1}{n_i} \sum_{j=1}^{n_i} l[g(\boldsymbol{x}_{i,j}), y^-].$$

Theoretically the selection of $\lambda_i^+$ and $\lambda_i^-$ is determined by constrains outlined in Eq. (9) (5) and (8). In practice they can be effectively computed by solving the following least squares problems. The determination of $\lambda_i^+$ is based on the following optimization problem:

$$\text{minimize} \quad \sum_{i=1}^{m} (\mu_i^+)^2, \tag{12a}$$

$$\text{subject to} \quad \begin{cases} \sum_{i=1}^{m} \mu_i^+ \sqrt{n_i} = \frac{1}{2}, \\ \sum_{i=1}^{m} \mu_i^+ \pi_i \sqrt{n_i} = \frac{1}{2}, \end{cases} \tag{12b}$$

$$\text{obtain} \quad \lambda_i^+ = \mu_i^+ \sqrt{n_i}. \tag{12c}$$

Similarly, $\lambda_i^-$ is determined by the following optimization problem:

$$\text{minimize} \quad \sum_{i=1}^{m} (\mu_i^-)^2, \tag{13a}$$

$$\text{subject to} \quad \begin{cases} \sum_{i=1}^{m} \mu_i^- \sqrt{n_i} = \frac{1}{2}, \\ \sum_{i=1}^{m} \mu_i^- \pi_i \sqrt{n_i} = 0, \end{cases} \tag{13b}$$

$$\text{obtain} \quad \lambda_i^- = \mu_i^- \sqrt{n_i}. \tag{13c}$$

## C    PROOF OF THEOREMS

### C.1    PROOF OF THEOREM 1

**True risks:** In both Eq. (1) and (2), the absolute risks are expressed as linear combinations of true risks:

$$\mathcal{R}_i^+ = \pi_i \mathcal{R}_+^+ + (1 - \pi_i)(1 - \mathcal{R}_-^-),$$
$$\mathcal{R}_i^- = \pi_i (1 - \mathcal{R}_+^+) + (1 - \pi_i)\mathcal{R}_-^-, \tag{14}$$

where $\pi_i$ is the ground truth class posterior of the $i$th subset. Combining the Eqs. (14) and (2), we can express the risk estimator as a linear combination of true risks:

$$\mathcal{R}^+_{postPU} = \mathcal{R}^+_+ \sum_{i=1}^m \lambda_i^+ \pi_i - \mathcal{R}^-_- \sum_{i=1}^m \lambda_i^+ (1 - \pi_i) + \sum_{i=1}^m \lambda_i^+ (1 - \pi_i), \tag{15a}$$

$$\mathcal{R}^-_{postPU} = -\mathcal{R}^+_+ \sum_{i=1}^m \lambda_i^- \pi_i + \mathcal{R}^-_- \sum_{i=1}^m \lambda_i^- (1 - \pi_i) + \sum_{i=1}^m \lambda_i^- \pi_i, \tag{15b}$$

A linear transformation $f(\mathcal{R}_2) = \frac{k_2^+}{k_1^+}(\mathcal{R}_1 - c_1) + c_2$ can be established from $\mathcal{R}_1$ to $\mathcal{R}_2$, where $\frac{k_2^+}{k_1^+} = \frac{k_2^-}{k_1^-} > 0$. Then $\mathcal{R}_1(g_1^*, p) = \min(\mathcal{R}_1(g, p))$ implies $\mathcal{R}_2(g_1^*, p) = f[\mathcal{R}_1(g_1^*, p)] = \min_{g \in \mathcal{G}} f[\mathcal{R}_1(g_1^*, p)] = \min_{g \in \mathcal{G}} \mathcal{R}_2(g, p)$. Similarly we can prove $\mathcal{R}_1(g_2^*, p) = \min \mathcal{R}_1(g, p)$.

Notably, when the estimated class prior probability is less than half of the true class prior probability, a potential hazard arises. Specifically, if $\hat{\pi} < \pi/2$ it causes $k^+$Plessis et al. (2014) to become negative. This, in turn, leads to the reversal of the gradient descent direction and results in a trivial classifier that outputs only the positive label. Selecting sample weights that satisfy Eq. (5) can also address this issue. This observation is verified by the zero F1 scores obtained for the positive class when the classifiers trained by nnPU are subjected to the condition $\hat{\pi} < 0.5\pi$ (red dashed line in Fig. 1(d) ) .

## C.2 Proof of Theorem 2

For the sufficient condition, we can calculate the derivatives of $\hat{\bar{\pi}}$ from the definition in Eq.(4):

$$\frac{\partial \hat{\bar{\pi}}}{\partial \pi_j} = -\frac{(\lambda_j^+ - \lambda_j^-) \sum_{i=1}^m (\lambda_i^+ - \lambda_i^-)}{\left[\sum_{i=1}^m (2\pi_i - 1)(\lambda_i^+ - \lambda_i^-)\right]^2} \tag{16}$$

Note that $\forall \boldsymbol{\pi} \in D, \hat{\bar{\pi}}(\boldsymbol{\pi}) = \hat{\bar{\pi}}(\hat{\boldsymbol{\pi}})$ implies $\forall \boldsymbol{\pi} \in D, j = 1, 2, \ldots, m, \partial \hat{\bar{\pi}}(\boldsymbol{\pi})/\partial \pi_j = 0$. Equating the right-hand side of Eq.(16) for all $j$, we have $\sum_{i=1}^m \lambda_i^+ - \lambda_i^- = 0$.

For the necessary condition, let $\boldsymbol{v} = (\lambda_1^+ - \lambda_1^-, \lambda_2^+ - \lambda_2^-, \ldots, \lambda_m^+ - \lambda_m^-)$, then we obtain true class coefficient $k^+, k^-$ in terms of inner products:

$$k^+(\boldsymbol{\pi}) = \sum_{i=1}^m (\lambda_i^+ - \lambda_i^-)\pi_i = \langle \boldsymbol{v}, \boldsymbol{\pi} \rangle,$$

$$k^-(\boldsymbol{\pi}) = \sum_{i=1}^m (\lambda_i^+ - \lambda_i^-)(\pi_i - 1) = \langle \boldsymbol{v}, \boldsymbol{\pi} \rangle - \langle \boldsymbol{v}, \mathbf{1} \rangle. \tag{17}$$

Since $\sum_{i=1}^m \hat{\lambda_i^+} - \hat{\lambda_i^-} = 0 \iff \langle \boldsymbol{v}, \mathbf{1} \rangle = 0$, we have $k^+ = k^-$. Since $\boldsymbol{\pi} \in D$ ensures $\langle \boldsymbol{\pi}, \mathbf{1} \rangle \neq 0$, we combine $\forall \boldsymbol{\pi} \in D, k^+(\boldsymbol{\pi}) = k^-(\boldsymbol{\pi})$ with $\bar{\pi} = k^+/(k^+ + k^-)$ and obtain $\forall \boldsymbol{\pi} \in D, \hat{\bar{\pi}}(\boldsymbol{\pi}) = \frac{1}{2} = \hat{\bar{\pi}}(\hat{\boldsymbol{\pi}})$.

## D ANALYTICAL SOLUTIONS

### D.1 DETERMINATION OF SAMPLE WEIGHTS

In 2-subset case the choice of $\lambda_i^+$ and $\lambda_i^-$ are:

$$\lambda_1^+ = \frac{1 - \pi_2}{2(\pi_1 - \pi_2)}, \tag{18a}$$

$$\lambda_2^+ = \frac{1 - \pi_1}{2(\pi_2 - \pi_1)}, \tag{18b}$$

$$\lambda_1^- = -\frac{\pi_2}{2(\pi_1 - \pi_2)}, \tag{18c}$$

$$\lambda_2^- = -\frac{\pi_1}{2(\pi_2 - \pi_1)}. \tag{18d}$$

In traditional PU learning setting where $\pi_1 = 1, \pi_2 = \pi$ we have:

$$\lambda_1^+ = \frac{1}{2}, \tag{19a}$$

$$\lambda_2^+ = 0, \tag{19b}$$

$$\lambda_1^- = -\frac{\pi}{2(1 - \pi)}, \tag{19c}$$

$$\lambda_2^- = \frac{1}{2(1 - \pi)}. \tag{19d}$$

In 3-subset case the choice of $\lambda_i^+$ and $\lambda_i^-$ are:

$$\lambda_1^+ = \frac{n_1[n_2(1 - \pi_2)(\pi_1 - \pi_2) + n_3(1 - \pi_3)(\pi_1 - \pi_3)]}{2[n_1 n_2(\pi_1 - \pi_2)^2 + n_1 n_3(\pi_1 - \pi_3)^2 + n_2 n_3(\pi_2 - \pi_3)^2]}, \tag{20a}$$

$$\lambda_2^+ = \frac{n_2[n_1(1 - \pi_1)(\pi_2 - \pi_1) + n_3(1 - \pi_3)(\pi_2 - \pi_3)]}{2[n_1 n_2(\pi_1 - \pi_2)^2 + n_1 n_3(\pi_1 - \pi_3)^2 + n_2 n_3(\pi_2 - \pi_3)^2]}, \tag{20b}$$

$$\lambda_3^+ = \frac{n_3[n_1(1 - \pi_1)(\pi_3 - \pi_1) + n_2(1 - \pi_2)(\pi_3 - \pi_2)]}{2[n_1 n_2(\pi_1 - \pi_2)^2 + n_1 n_3(\pi_1 - \pi_3)^2 + n_2 n_3(\pi_2 - \pi_3)^2]}, \tag{20c}$$

$$\lambda_1^- = \frac{n_1[n_2 \pi_2(\pi_2 - \pi_1) + n_3 \pi_3(\pi_3 - \pi_1)]}{2[n_1 n_2(\pi_1 - \pi_2)^2 + n_1 n_3(\pi_1 - \pi_3)^2 + n_2 n_3(\pi_2 - \pi_3)^2]}, \tag{20d}$$

$$\lambda_2^- = \frac{n_2[n_1 \pi_1(\pi_1 - \pi_2) + n_3 \pi_3(\pi_3 - \pi_2)]}{2[n_1 n_2(\pi_1 - \pi_2)^2 + n_1 n_3(\pi_1 - \pi_3)^2 + n_2 n_3(\pi_2 - \pi_3)^2]}, \tag{20e}$$

$$\lambda_3^- = \frac{n_3[n_1 \pi_1(\pi_1 - \pi_3) + n_2 \pi_2(\pi_2 - \pi_3)]}{2[n_1 n_2(\pi_1 - \pi_2)^2 + n_1 n_3(\pi_1 - \pi_3)^2 + n_2 n_3(\pi_2 - \pi_3)^2]}. \tag{20f}$$

$$\tag{20g}$$

### D.2 OPTIMAL EFFECTIVE SAMPLE SIZES

In 2-subset case the optimal effective sample size is:

$$\frac{4 n_1 n_2(\pi_1 - \pi_2)^2}{n_1[\pi_1^2 + (1 - \pi_1)^2] + n_2[\pi_2^2 + (1 - \pi_2)^2]} \tag{21}$$

Corresponding effective sample size for nnPU ($\pi_1 = 1.0$) is:

$$\frac{n_1 n_2}{n_1 + n_2[\pi_2^2 + (1 - \pi_2)^2]} \tag{22}$$

In 3-subset case the optimal effective sample size is:

$$\frac{4[n_1 n_2(\pi_1 - \pi_2)^2 + n_1 n_3(\pi_1 - \pi_3)^2 + n_2 n_3(\pi_2 - \pi_3)^2]}{n_1[\pi_1^2 + (1 - \pi_1)^2] + n_2[\pi_2^2 + (1 - \pi_2)^2] + n_3[\pi_3^2 + (1 - \pi_3)^2]} \tag{23}$$

# E    GENERALIZATION ERROR BOUNDS

Suppose we aim to find the optimal classifier $g^*$ within function space $\mathcal{G}$ that minimizes the expected risk $\mathcal{R}(g)$. We denote the expected risk by $\mathcal{R}(g) = 1/2[\mathcal{R}_+^+(g) + \mathcal{R}_-^-(g)] = 1/2\mathbb{E}_{p(x|y+)}l[g(\boldsymbol{x}), y+] + 1/2\mathbb{E}_{p(x|y-)}l[g(\boldsymbol{x}), y-]$. And the empirical risk corresponds to the empirical version of postPU risk estimator $\hat{\mathcal{R}}(g) = \hat{\mathcal{R}}_{postPU}(g)$. Let $\hat{g}$ be the classifier that minimizes the empirical risk $\hat{\mathcal{R}}(g)$. Let $\mathfrak{R}_{n_i, V_i}(\mathcal{G})$ denote the Rademacher complexityMehryar Mohri & Talwalkar (2012) of $\mathcal{G}$ for the sampling of size $n_i$ with probability $P(x|V_i)$, where $V_i$ is the evidence corresponding to subset $\mathcal{X}_i$ of the training set.

**Lemma 1.** *Following the above assumptions, for any $\delta > 0$ with probability at least $1 - \delta$,*

$$\sup_{g \in \mathcal{G}} \left| \hat{\mathcal{R}}_i^+(g) - \mathcal{R}^+(g) \right| \leq 2\phi_g \mathfrak{R}_{n_i, V_i}(\mathcal{G}) + \sqrt{\frac{\ln(4/\delta)}{2n_i}} \tag{24}$$

$$\sup_{g \in \mathcal{G}} \left| \hat{\mathcal{R}}_i^-(g) - \mathcal{R}^-(g) \right| \leq 2\phi_g \mathfrak{R}_{n_i, V_i}(\mathcal{G}) + \sqrt{\frac{\ln(4/\delta)}{2n_i}} \tag{25}$$

This lemma is proven by employing *McDiarmid's inequality* and uniform deviation bounds using the Rademacher complexityMehryar Mohri & Talwalkar (2012). It is analogous with Lemma 8 and Eq. (17) in Niu et al. (2016).

**Lemma 2.**

$$\hat{\mathcal{R}}_{postPU}(g) - \mathcal{R}(g) = \sum_{i=1}^m \lambda_i^+ [\hat{\mathcal{R}}_i^+(g) - \mathcal{R}^+(g)] + \sum_{i=1}^m \lambda_i^- [\hat{\mathcal{R}}_i^-(g) - \mathcal{R}^-(g)] \tag{26}$$

This lemma is proven through decomposition:

$$\hat{\mathcal{R}}_{postPU}(g) - \mathcal{R}(g)$$

$$= \hat{\mathcal{R}}_{postPU}^+(g) + \hat{\mathcal{R}}_{postPU}^-(g) - \mathcal{R}_+^+(g)/2 - \mathcal{R}_-^-(g)/2$$

$$= \sum_{i=1}^m \lambda_i^+ \hat{\mathcal{R}}_i^+(g) + \lambda_i^- \hat{\mathcal{R}}_i^-(g) + \sum_{i=1}^m \lambda_i^+ \pi_i \mathcal{R}_+^+(g) + \lambda_i^- (1 - \pi_i)\mathcal{R}_-^-(g) \tag{27a}$$

$$= \sum_{i=1}^m \lambda_i^+ \hat{\mathcal{R}}_i^+(g) + \lambda_i^- \hat{\mathcal{R}}_i^-(g) - \sum_{i=1}^m \lambda_i^+ \pi_i \mathcal{R}_+^+(g) - \lambda_i^+ (1 - \pi_i)\mathcal{R}_+^+(g)$$

$$\quad - \sum_{i=1}^m \lambda_i^- (1 - \pi_i)\mathcal{R}_-^-(g) - \lambda_i^- \pi_i \mathcal{R}_+^-(g) \tag{27b}$$

$$= \sum_{i=1}^m \lambda_i^+ \hat{\mathcal{R}}_i^+(g) + \lambda_i^- \hat{\mathcal{R}}_i^-(g) - \sum_{i=1}^m \lambda_i^+ \mathcal{R}^+(g) - \sum_{i=1}^m \lambda_i^- \mathcal{R}^-(g)$$

$$= \sum_{i=1}^m \lambda_i^+ [\hat{\mathcal{R}}_i^+(g) - \mathcal{R}^+(g)] + \sum_{i=1}^m \lambda_i^- [\hat{\mathcal{R}}_i^-(g) - \mathcal{R}^-(g)].$$

Eq. (27a) is obtained by applying Eq. (5) and Eq. (27b) is obtained by applying Eq. (9).

Finally Theorem 4 is proven through:

$$\mathcal{R}(\hat{g}) - \mathcal{R}(g^*)$$

$$= [\hat{\mathcal{R}}_{postPU}(\hat{g}) - \hat{\mathcal{R}}_{postPU}(g^*)] + [\mathcal{R}(\hat{g}) - \hat{\mathcal{R}}_{postPU}(\hat{g})] + [\hat{\mathcal{R}}_{postPU}(g^*) - \mathcal{R}(g^*)]$$

$$\leq 0 + 2\sup_{g \in \mathcal{G}} \left| \hat{\mathcal{R}}_{postPU}(g) - \mathcal{R}(g) \right|$$

$$\leq 2\sum_{i=1}^m \lambda_i^+ \sup_{g \in \mathcal{G}}[\hat{\mathcal{R}}_i^+(g) - \mathcal{R}^+(g)] + 2\sum_{i=1}^m \lambda_i^- \sup_{g \in \mathcal{G}}[\hat{\mathcal{R}}_i^-(g) - \mathcal{R}^-(g)] \tag{28a}$$

$$\leq \sum_{i=1}^m 4(\lambda_i^+ + \lambda_i^-)\phi_g \mathfrak{R}_{n_i, V_i}(\mathcal{G}) + \sum_{i=1}^m (\lambda_i^+ + \lambda_i^-)\sqrt{\frac{2\ln(4/\delta)}{n_i}} \tag{28b}$$

Eq. (28a) is obtained by employing Lemma 2 and Eq. (10) is obtained by employing Lemma 1.

An asymptotic analysis of the computational cost is presented below: (1) Initialization stage: Sample weights $\lambda_i^+, \lambda_i^-$ are obtained by solving the linear optimization in Eq. (8) via least square minimization. The dominant operation involves SVD of a $2m \times 2$ matrix, resulting in $\mathcal{O}(m)$ complexity. This initialization procedure is executed only once. (2) Training stage: In each epoch, the empirical risk is computed using Eq.(2), which involves calculating linear combinations of sample-wise surrogate losses weighted by the precomputed coefficients $\lambda_i^+, \lambda_i^-$. This process maintains a computational complexity of $\mathcal{O}(n)$, consistent with conventional nnPU methods.

## F    SPECIFICATION OF EXPERIMENTS

**Data preparation:** Both datasets originally consist of 10 class labels ranging from 0 to 9. In our study, we reclassify these labels by designating classes 0 through 6 as positive (P) and classes 7 through 9 as negative (N). Subsequently, we then generate two exclusive training sets: a positive set $\mathcal{X}_P$ and an unlabeled set $\mathcal{X}_U$ with a positive class probability of $\pi_p = 40\%$. The positive set samples are selected randomly, while the test set samples are independent of the training sets. Detailed statistics for each dataset are provided in Table 5.

Table 5: Specification of experiment setting on each dataset.

| DataSet | Unlabeled training set | | Positive training set | Test set | |
|---------|------------------------|----------|------------------------|----------|-----------|
|         | samples | P ratio(%) | samples | samples | P ratio(%) |
| CIFAR-10 | 25,000 | 40 | 204 | 5,000 | 40 |
| F-MNIST | 30,000 | 40 | 631 | 5,000 | 40 |

This experimental setup simulates a semi-supervised learning scenario in which we possess a small but accurately labeled positive set $\mathcal{X}_P$ and a larger unlabeled set $\mathcal{X}_U$. In addition, we assume the existence of an automatic, yet imperfect, labeling method that identifies all samples that meet a specific criterion $V$. Among these identified samples, only $P(y = y^+ | V \wedge \boldsymbol{x} \in \mathcal{X}_U) = \pi_1 = 50\%$ are positive. We denote by $r = P(V | y = y^+ \wedge \boldsymbol{x} \in \mathcal{X}_U)$ the proportion of true positives that are labeled based on evidence $V$. Posterior probabilities allow for a more precise characterization of $\mathcal{X}_U$ using the evidence $V$, as opposed to relying solely on an overall prior probability.

**Baselines:** We compare our proposed method with 5 state-of-the-art PU learning baselines and a supervised baseline. Among these, two are native empirical risk minimization (ERM) methods. Including the supervised baseline and the proposed postPU method, these methods directly optimize an estimated risk via gradient descent.

nnPUKiryo et al. (2017): A commonly used PU method that follows a general ERM framework, directly utilizing a non-negative PU risk estimator as the overall loss function. It often serves as a benchmark for PU learning methods.

ImbPUSu et al. (2021): An augmentation of nnPUKiryo et al. (2017) which introduces class balancing coefficients to the PU risk estimator.

The remaining three comparison methods involve augmentations beyond the general ERM framework, such as meta-learning or self-learning techniques applied to nnPU or the supervised method. It is worth noting that some of these augmentations can be combined with any arbitrary native ERM method, including our proposed postPU method.

LaGAMLong et al. (2024): A meta-learning PU method that enables more aggressive label disambiguation through contrastive learning.

robustPUZhu et al. (2023): A PU method that distinguishes likely negatives and dynamically updates sample weights in a curriculum learning framework.

DistPUJiang et al. (2023): A PU method that aligns the label distribution between the predictions and the ground-truth labels.

**Implementation Details:** We use the open-source code provided by the authors of these methods. For a fair comparison, we adopt the same architecture for classifier $\mathcal{G}$ across all comparing methods. Specifically, $\mathcal{G}$ consists of 2 convolutional layers (CNN) followed by 2 fully connected layers with ReLU activation. Throughout the experiments, we maintain a consistent learning rate of $1 \times 10^{-4}$, a uniform batch size of 2048, and a maximum of 100 epoch. Each experimental case is executed over the course of 10 trials to ensure statistical reliability.

**Detailed results** Quantitative results of experiment on CIFAR-10 and F-MNIST dataset is reported in Table 6.

Table 6: Quantitative results of PU learning experiment on (A) CIFAR-10 and (B) F-MNIST dataset.

|  | Metrics(%) | postPU | ImbPU | nnPU | LaGAM | robustPU | DistPU |
|---|---|---|---|---|---|---|---|
| A | F1 | $64.4 \pm 0.5$ | $55.2 \pm 1.9$ | $34.5 \pm 7.5$ | $57.0 \pm 2.3$ | $55.4 \pm 0.2$ | $49.1 \pm 2.6$ |
|  | accuracy | $67.0 \pm 0.4$ | $66.1 \pm 0.5$ | $66.0 \pm 1.2$ | $65.3 \pm 1.5$ | $71.5 \pm 0.0$ | $64.5 \pm 0.4$ |
|  | precision | $56.7 \pm 0.7$ | $58.6 \pm 1.2$ | $75.7 \pm 4.0$ | $56.5 \pm 1.9$ | $73.9 \pm 0.2$ | $57.5 \pm 0.4$ |
|  | recall | $74.7 \pm 2.4$ | $52.5 \pm 4.4$ | $22.9 \pm 6.4$ | $57.7 \pm 4.1$ | $44.3 \pm 0.3$ | $42.9 \pm 4.2$ |
| B | F1 | $94.7 \pm 0.1$ | $94.4 \pm 0.1$ | $93.5 \pm 0.1$ | $79.7 \pm 6.5$ | $88.7 \pm 0.1$ | $91.7 \pm 0.3$ |
|  | accuracy | $95.8 \pm 0.1$ | $95.6 \pm 0.0$ | $95.0 \pm 0.1$ | $84.9 \pm 5.1$ | $91.4 \pm 0.0$ | $93.2 \pm 0.3$ |
|  | precision | $95.1 \pm 0.5$ | $96.1 \pm 0.2$ | $96.3 \pm 0.5$ | $86.9 \pm 9.2$ | $94.1 \pm 0.2$ | $89.1 \pm 0.9$ |
|  | recall | $94.3 \pm 0.4$ | $92.8 \pm 0.2$ | $90.9 \pm 0.6$ | $74.0 \pm 6.5$ | $83.8 \pm 0.3$ | $94.4 \pm 0.4$ |

# G  APPLICATION IN INDUSTRIAL ANOMALY DETECTION

In data-driven anomaly detection, a critical challenge arises from the inherent inconsistency between anomalies (the detection target) and failure records (labels for model training). Early-stage anomalies minimally impact operational metrics (e.g., wind turbine power), rendering performance-based monitoring systems ineffective. Limited sensor coverage further necessitates manual inspections, yielding sparse, delayed, and temporally imprecise anomaly documentation. Consequently, direct accurate labels are critically scarce. Compensatory use of failure records introduces discrepancies , exacerbating annotation uncertainty and inaccuracies—particularly for early-stage detection.

This experiment employs a public operational dataset from a 3 MW Irish wind turbine Leahy et al. (2016). The dataset comprises: (a) SCADA records spanning January 2014 to December 2016 (49,027 timestamps). (b) Event logs documenting 213 critical faults. During preprocessing, records corresponding to non-operational states were removed, followed by data normalization and segmentation into one-hour intervals. To mitigate class imbalance due to rare anomalous samples, we performed data augmentation using the Synthetic Minority Over-sampling Technique (SMOTE). The dataset was partitioned into training (January 5–October 20, 2014) and testing (October 20–December 31, 2014) subsets. Each case was evaluated over 10 iterations to ensure statistical reliability.

Specifically, $\pi_1$ is estimated based on the time range $D$, calculated as $\pi_l = P(t \notin (\tau_i^* - D_w, \tau_i^* + D_w)|t \in (\hat{\tau}_i - D, \hat{\tau}_i)) = (D - D_w)^2/D^2$. For traditional PU methods requiring homogeneous unlabeled sets, anomalous and likely-anomalous samples are combined with a unified prior probability.

Table 7: Summary of general notations with definitions

| Notation | Definition |
|---|---|
| $\boldsymbol{x}, y$ | Input features $\boldsymbol{x} \in \mathcal{R}^d$; class label $y \in \{y^-, y^+\}$ |
| $p$ | True probability distribution |
| $g$ | Classification model $\hat{y} = g(x)$ |
| $l$ | Loss function $l[\hat{y}, y]$ |
| $\pi$ | True prior probability $P(y = y^+)$ |
| $\pi_i$ | True posterior probability $P(y = y^+ \| V_i)$ given observed evidence $V_i$ |
| $\hat{\pi}_i$ | Estimated/given probability (inaccurate) |
| $\delta$ | Probability estimation error $\hat{\pi}_i - \pi_i$ |
| $\overline{r}$ | Effective class weight, the metric proposed in Theorem 1. |
| $\mathcal{X}_i$ | Training samples with evidence $V_i$ observed |
| $m$ | Total number of subsets $\mathcal{X}_i$ |
| $n_i$ | Size of subset $\mathcal{X}_i$ |
| $\mathcal{R}$ | Expected risk, learning objective |
| $\mathcal{R}_i^+, \mathcal{R}_i^-$ | True positive/negative risk on $\mathcal{X}_i$ |
| $\lambda_i^+, \lambda_i^-$ | Sample weights corresponding to $y^+, y^-$ on $\mathcal{X}_i$ |

