# OpenReview forum: "Positive and unlabeled learning incorporating additional posterior probabilities"
_ICLR.cc/2026/Conference — Submitted to ICLR 2026_

### Official Review · Reviewer_Lyij · 2025-10-27

**Soundness:** 3
**Presentation:** 3
**Contribution:** 3
**Rating:** 6
**Confidence:** 3

**Summary:**

Although conventional PU learning methods often assume a known class prior, this assumption is unrealistic, and the associated hyperparameter is highly sensitive. To address this, the paper proposes a generalized framework that assigns a posterior probability to each data subset in PU learning. By using estimable per-subset sample weights $\lambda^{+}$ and $\lambda^{-}$, the framework enables PU learning that is both robust to errors in the posterior probabilities and high-performing. The effectiveness of the proposed method is demonstrated theoretically and empirically.

**Strengths:**

- Once $\lambda^{+}$ and $\lambda^{-}$ are estimated, the procedure closely follows nnPU, making the proposed method simple yet effective.
- The method theoretically and experimentally exhibits strong robustness and its performance surpasses existing approaches.
- The explanation uses figures and tables effectively, making the presentation visually clear and easy to understand.

**Weaknesses:**

- I have several questions; please see Questions.
- Minor comments: There appear to be many places where \citet and \citep are used incorrectly.

**Questions:**

- How does the proposed method relate to class-prior-free PU learning approaches such as [1, 2] ? If possible, a quantitative comparison would be valuable.
- This paper primarily reports F1-score as the evaluation metric. Is there a reason for not using AUC? Especially for anomaly detection, AUC might be more appropriate.

[1] Chen, Hui, et al. "A variational approach for learning from positive and unlabeled data." Advances in Neural Information Processing Systems 33 (2020): 14844-14854.

[2] Zhao, Hengwei, et al. "Class prior-free positive-unlabeled learning with Taylor variational loss for hyperspectral remote sensing imagery." Proceedings of the IEEE/CVF International Conference on Computer Vision. 2023.

---

> ### Author Response · Authors · 2025-11-24
> **Response to Reviewer Lyij**
>
> We appreciate the reviewer's feedback and provide the following responses and new results:
>
> **Q1: Relationship to Class-Prior-Free PU Learning Methods**
>
> **A1:** We thank the reviewer for this insightful question regarding the relationship between our method and class-prior-free PU learning approaches. While both paradigms address challenges in PU learning, they operate under different assumptions and mechanisms:
>
> **Conceptual Distinction:** Typical class-prior-free methods rely on explicit prior estimation, whereas our postPU framework utilizes coarsely approximated posterior probabilities across data subsA central advantage of our approach is its robustness to inaccuracies in these probability estimates, as theoretically guaranteed by **Theorem 3**.
>
> **Theoretical Foundation:** Theorem 3 establishes that our risk estimator remains unbiased even under substantial deviations of the estimated posteriors $\hat{\pi}_i$ from the true $\pi_i$, provided the condition
>
> $\sum_{i=1}^{m}(\lambda^+_i-\lambda^-_i)=0$
>
> is met. This theoretical assurance allows effective learning using only roughly approximated posteriors, a conclusion further supported by the empirical results in Section 6.2.
>
> **Empirical Comparison:** To enable quantitative comparison, we have incorporated results from a class-prior-free method HolisticPU [1], as summarized below:
>
> **Table: Comparison with Class-Prior-Free Method HolisticPU**
> | Dataset | Method | Accuracy | Precision | Recall | F1-Score |
> |---|---|---|---|---|---|
> | Cifar10 | HolisticPU | 40.8 ± 0.5 | 40.0 ± 0.1 | 95.5 ± 3.5 | 56.3 ± 0.7 |
> | Fmnist | HolisticPU | 79.87 ± 0.0 | 88.86 ± 0.0 | 75.99 ± 0.0 | 81.92 ± 0.0 |
>
> [1] Wang, Xinrui, et al. "Beyond Myopia: Learning from Positive and Unlabeled Data through Holistic Predictive Trends." *Proceedings of the 37th International Conference on Neural Information Processing Systems* (2023): 2955.
>
> **Q2: Evaluation Metrics - F1-Score vs. AUC**
>
> **A2:** We agree with the reviewer that AUC offers valuable complementary insights. In response, we have expanded our evaluation to include AUC comparisons across all methods, as presented below:
>
> **Table: AUC Comparison (%) Across Methods**
> | Dataset | postPU | ImbPU | nnPU | Supervised | LaGAM | RobustPU | DistPU |
> |---|---|---|---|---|---|---|---|
> | Cifar10 | 68.3 ± 0.1 | 63.8 ± 0.6 | 58.9 ± 2.1 | 50.0 ± 0.0 | 64.0 ± 1.6 | 66.9 ± 0.1 | 60.9 ± 1.0 |
> | Fmnist | 95.5 ± 0.1 | 95.1 ± 0.1 | 94.3 ± 0.2 | 50.0 ± 0.0 | 83.1 ± 5.1 | 90.1 ± 0.0 | 93.4 ± 0.2 |
>
> **Q3: Citation Formatting**
>
> **A3:** We thank the reviewer for noting the citation formatting issues. We have thoroughly reviewed the manuscript and corrected all instances of `\citet` and `\citep` usage to ensure proper distinction between textual and parenthetical citations throughout the paper.

---

> > ### Comment · Reviewer_Lyij · 2025-11-27
> >
> > Thank you for your response. I will keep the score at 6, but I will increase my confidence.

---

### Official Review · Reviewer_LJUZ · 2025-10-31

**Soundness:** 2
**Presentation:** 3
**Contribution:** 2
**Rating:** 4
**Confidence:** 4

**Summary:**

To address the dual challenges of inaccurate class prior estimation and limited representation in training labels, this paper proposes a posterior-based PU learning framework (postPU) that extends traditional prior-based PU learning by modeling uncertainty through subset-specific posterior probabilities. The authors introduce a class-balanced weighting principle to enhance robustness against class probability estimation errors and provide theoretical guarantees through generalization error bounds. Experiments on CIFAR-10, F-MNIST, and wind turbine anomaly detection demonstrate the method's effectiveness in leveraging auxiliary uncertain annotations.

**Strengths:**

(1) The extension from prior-based to posterior-based PU learning provides a more flexible formulation that can naturally incorporate uncertain annotations from multiple sources with different confidence levels.
(2) The potential for real-world scenarios is demonstrated through the wind turbine anomaly detection application.
(3) The paper is overall well written with strong theoretical contributions.

**Weaknesses:**

(1) The paper's motivation centers on addressing inaccurate prior estimation in traditional PU learning. However, the proposed solution pivots to requiring posterior probability estimates for each subset. While the method claims robustness to "coarsely approximated posterior values", the robustness demonstration in Table 4 still requires manual specification of posterior values, which test sensitivity to posterior estimation errors, but the baseline "accurate posterior probabilities" themselves require strong assumptions about the data generating process that may not hold. The experimental setup appears largely self-fulfilling.
(2) In lines 837-843, the paper describes constructing a semi-supervised environment using a heuristic criterion V that identifies samples meeting a specific criterion. But how is criterion V defined or learned? How are the posterior probabilities computed or estimated for each subset? Why should practitioners have access to reliable posterior estimates when prior estimation is already challenging?
(3) Several critical experiments are missing:
(a) The theoretical framework supports arbitrary $m \ge 2$ subsets, but experiments only evaluate $m=2$ and $m=3$ cases. Lack of experiments with $m>3$ leaves the scalability claims empirically unsupported.
(b) The paper lacks the hyperparameter sensitivity analysis on $\beta$.
(c) The comparison lacks several relevant recent PU learning methods, such as HolisticPU, PUL-CPBF, RobustPU and other methods from 2023-2025.

**Questions:**

Please refer to the Weakness.

---

> ### Author Response · Authors · 2025-11-24
> **Response to Reviewer LJUZ**
>
> We thank the reviewer for these insightful comments and constructive criticisms. Below, we provide a point-by-point response to address the concerns raised.
>
> **Q1: Self-fulfilling experimental setup for robustness validation**
>
> **A1:** We apologize for the misuse of term "baseline" in our original description, which may have led to a misunderstanding.
>
> **Clarification of the "Oracle Case":** The case with "accurate posterior probabilities" in our experiments was not intended as a "baseline" but rather as an **"oracle case"**—an reference point used to illustrate an ideal scenario.
>
> **Intent of the Experiment:** The primary purpose of this experiment is to provide an **empirical validation of Theorem 3**. It demonstrates that our risk estimator remains robust and performs stably even when the provided posterior values deviate significantly from the actual values. The performance in these cases is comparable to the oracle case, underscoring that the risk estimator is resistant to inaccuracies in the class probability estimates. It illustrates a practical application of our method. Specifically, it only requires a coarse indication of which subset has a higher probability of containing positive samples, rather than precise posterior probability values.
>
> **Q2: Definition of Criterion V and Feasibility of Posterior Estimates**
>
> **A2:** We thank the reviewer for this question, which allows us to elaborate on the practical applicability of our framework.
>
> **Nature of Criterion V:** The criterion V is defined based on **heuristic prior knowledge** about which parts of the training data are more (or less) likely to contain positive samples. We argue that inferring such **relative relationship between subsets** is often more feasible in practice than estimating a precise global prior for the entire dataset.
>
> **Simulated Environment and Real-world Scenarios:** The semi-supervised setup models real-world cases where such imprecise prior knowledge is available. For instance: In semi-supervised medical diagnosis, practitioners might know that elderly patients have a higher probability of a certain disease; in spam filtering, emails containing keywords like "prize" or "winner" constitute a subset with a higher likelihood of being spam.
>
> The proposed method does not require accurate posterior estimation (as supported by Theorem 3) but only the **relative order** of positive sample probabilities across subsets. We are currently supplementing the manuscript with more meaningful experimental settings for V.
>
> **Q3: Missing Critical Experiments**
>
> **Q3(a) scalability validation for m>3**
>
> **A3(a):** We have conducted additional experiments to assess the computational overhead for larger values of m (number of subsets) and n (number of samples). The results below confirm the expected O(mn) time complexity. Since m is typically much smaller than n (e.g., m < 10 in most practical cases), the dominant factor remains O(n), confirming the scalability of our approach.
>
> **Table: Computational Time (seconds) vs. Number of Subsets (m) and Samples (n)**
> | n \ m | 3 | 5 | 10 | 100 | 1000 | 10000 |
> |---|---|---|---|---|---|---|
> | 10,000 | 11.3 | 12.84 | 20.0 | 131.59 | 1036.49 | 10349.57 |
> | 100,000 | 83.75 | 106.78 | 165.55 | 1182.29 | 10005.07 | 100780.04 |
> | 1,000,000 | 846.0 | 1195.09 | 1989.88 | 12904.53 | - | - |
>
> **Q3(b) hyperparameter sensitivity analysis on $\beta$**
>
> **A3(b)** We performed a hyperparameter sensitivity analysis on  $\beta$ , which is a parameter inherited from the nnPU method. It controls the learning rate for correcting negative empirical risk. The results below indicate that the performance of our method is not sensitive to the value of  $\beta$.
> **Table: Hyperparameter Sensitivity Analysis on $\beta$**
> | $\beta$ | F1-score(%) | accuracy(%) | precision(%) | recall(%) |
> |---|---|---|---|---|
> | 0.001 | 64.2 ± 0.3 | 67.0 ± 0.6 | 56.8 ± 0.8 | 74.0 ± 2.1 |
> | 0.01 | 64.3 ± 0.4 | 67.0 ± 0.4 | 56.7 ± 0.6 | 74.5 ± 2.2 |
> | 0.1 | 64.6 ± 0.3 | 66.6 ± 0.6 | 56.2 ± 0.8 | 76.0 ± 1.9 |
> | 0.2 | 64.3 ± 0.4 | 67.1 ± 0.6 | 56.8 ± 0.9 | 74.1 ± 2.7 |
> | 0.5 | 64.3 ± 0.3 | 67.1 ± 0.4 | 56.9 ± 0.7 | 73.8 ± 1.8 |
>
> **Q3.(c) Comparison lacks several relevant recent PU learning methods**
>
> **A3.(c):** We thank the reviewer for this suggestion and have now included comparisons with additional recent PU learning methods, **HolisticPU** and **RobustPU**. The results are demonstrated below and will be incorporated into the revised manuscript.
>
> **Comparison with Recent PU Learning Methods**
> | Dataset | Method | Accuracy(%) | Precision(%) | Recall(%) | F1-Score(%) |
> |---|---|---|---|---|---|
> | Cifar10 | HolisticPU | 40.8 ± 0.5 | 40.0 ± 0.1 | 95.5 ± 3.5 | 56.3 ± 0.7 |
> | Cifar10 | RobustPU | 71.5 ± 0.0 | 73.9 ± 0.2 | 44.3 ± 0.3 | 55.4 ± 0.2 |
> | Fmnist | HolisticPU | 79.87 ± 0.0 | 88.86 ± 0.0 | 75.99 ± 0.0 | 81.92 ± 0.0 |
> | Fmnist | RobustPU | 91.4 ± 0.0 | 94.2 ± 0.0 | 83.5 ± 3.0 | 88.6 ± 0.0 |

---

### Official Review · Reviewer_hc3X · 2025-10-31

**Soundness:** 3
**Presentation:** 3
**Contribution:** 2
**Rating:** 6
**Confidence:** 3

**Summary:**

This paper presents a novel approach in PU learning by introducing postPU, which extends existing approaches to handle uncertainty through subset-specific posterior probabilities. The proposed postPU goes beyond the limitation of prior-based methods that assume the class probability within the unlabeled set is known. The authors provides solid theoretical analysis with convergence guarantees and generalization error bounds, and demonstrate enhanced robustness against existing methods.

**Strengths:**

1. This work relaxes the strict assumption of traditional PU learning, enables applications to more realistic scenarios with varying data distributions.
2. This work provides a principled approach to incorporate and handle uncertainty in training data.

**Weaknesses:**

While computational complexity is mentioned, there's insufficient analysis of practical scalability to large datasets or high-dimensional spaces. Runtime and memory comparisons with baseline methods are missing.
Experiments focus primarily on image datasets. The evaluation should include diverse domains like text classification, time series analysis, or graph data where PU learning is commonly applied to demonstrate broader applicability.

**Questions:**

See above weakness section.

---

> ### Author Response · Authors · 2025-11-24
> **Response to Reviewer hc3X**
>
> We appreciate the reviewer's feedback on computational efficiency and experimental scope. Our responses and new results are summarized below:
>
> **Q1: Insufficient Analysis of Practical Scalability**
>
> **A1:** We thank the reviewer for raising this important point. We have conducted additional experiments to thoroughly evaluate the computational complexity and scalability of our method.
>
> **Computational Complexity Analysis:** The table below demonstrates the computational time across varying m and n. As theoretically expected, our method exhibits O(mn) time complexity, where m is the number of subsets and n is the sample size. In practical applications, m is typically very small (e.g., m < 10), making O(n) the dominant factor. The results confirm the scalability of our approach.
>
> **Table: Computational Time (seconds) vs. Number of Subsets (m) and Samples (n)**
> | n \ m | 3 | 5 | 10 | 100 | 1000 | 10000 |
> |---|---|---|---|---|---|---|
> | 10,000 | 11.3 | 12.84 | 20.0 | 131.59 | 1036.49 | 10349.57 |
> | 100,000 | 83.75 | 106.78 | 165.55 | 1182.29 | 10005.07 | 100780.04 |
> | 1,000,000 | 846.0 | 1195.09 | 1989.88 | 12904.53 | - | - |
>
> **Runtime Comparison with Baselines:** We further evaluate the training time of the compraed methods. As shown below, our method maintains competitive efficiency compared to most empirical risk minimization (ERM)-based approaches, while being significantly faster than meta-learning methods:
>
> **Table: Training Time Comparison (MM:SS)**
> | Dataset | postPU | Supervised | nnPU | ImbPU | DistPU | LaGAM | RobustPU |
> |---|---|---|---|---|---|---|---|
> | Cifar10 | 00:54.4 | 00:50.1 | 00:54.2 | 00:53.5 | 01:16.7 | 16:15.9 | 02:10.7 |
> | Fmnist | 00:43.3 | 00:38.1 | 00:38.4 | 00:39.1 | 01:16.8 | 06:05.8 | 02:17.0 |
>
> **Memory Usage Considerations:**
>
> Memory consumption is largely governed by model architecture rather than the specific PU learning algorithm. When the same backbone network is used, memory usage remains comparable across all methods.
>
> **Q2: Limited Domain Diversity in Experiments**
>
> **A2:** We agree with the reviewer that demonstrating broader applicability is important. While our initial submission focused on image classification, we are actively expanding our experimental evaluation to include diverse domains where PU learning is commonly applied. For instance: **Medical Diagnosis**: Modeling scenarios where age groups serve as natural subsets, with elderly patients having higher prior probability for certain conditions; **Spam Filtering**: Utilizing keyword presence (e.g., "prize", "winner") as heuristic indicators for message subsets with elevated spam likelihood.

---

### Official Review · Reviewer_q5ga · 2025-11-01

**Soundness:** 2
**Presentation:** 2
**Contribution:** 2
**Rating:** 4
**Confidence:** 5

**Summary:**

This paper considers a generalized PU learning setting where the class probability within the unlabeled set is unknown. It extends the prior-based PU learning as posterior-based PU learning and represents label uncertainty within unlabeled set by multiple class posterior probabilities, leading to a posterior-based PU risk estimator. Besiders, it also introduces a class-balanced weighting principle to enchance the robustness to estimation errors.

**Strengths:**

1. The studied problem is important to the literature.

2. Extensive experiments validate the effectiveness of the proposed method.

**Weaknesses:**

1. This paper attempts to represent label uncertainty by multiple class posterior probabilities for different subsets. However, how to partition data to several subsets is unclear, and the authors introduce an “automatic, albeit imperfect, labeling method that identifies samples meeting a specific criterion” for partitioning data. But what is the specific criterion?

2. In the proposed postPU, the authors estimate PU risks by weighting absolute risks in each subset as shown in Eq. (2). Thus, the weights are important for postPU and also affect its robustness shown in the proposed Theorems. However, how to calculate these weights? Although Eq.(9) presents a primary constraint for them with the posterior probabilities $\pi_i$, it depends on known posterior probabilities $\pi_i$. But the posterior probabilities $\pi_i$ are unknown in practice.

3. Theorem presentation is difficult to parse; definitions of variables and assumptions should be made explicit before the statement, for example, Theorem 1.

**Questions:**

Please refer to Weaknesses.

---

> ### Author Response · Authors · 2025-11-24
> **Response to Reviewer q5ga**
>
> We thank the reviewer for your thoughtful comments, which help clarify key methodological aspects of our work. Below, we provide detailed responses to each point raised.
>
> **Q1:Data Partitioning and the Definition of Criterion V**
>
> **A1:** We appreciate the reviewer’s question regarding the data partitioning process and the specific criterion. The partitioning is guided by **heuristic prior knowledge**, which identifies subsets of the training data that are more (or less) likely to contain positive samples.
>
> **Real-World Instantiation**: In our wind turbine application, for instance, obtaining the prior probability of gearbox lubrication failure across the entire dataset is difficult. However, we can define $\pi_i$ based on the conditional probability of failure within a specific time window, such as the 6 hour period preceding a fault, which can be reasonably estimated from historical data or domain knowledge. This method naturally segments the data into distinct subsets (e.g., at failure, pre-failure, and normal operation), each associated with significantly different likelihoods of containing positive instances.
>
> **General Applicability**: This semi-supervised framework is designed to emulate real‑world scenarios in which auxiliary, though imprecise, prior knowledge is accessible. For instance: Medical diagnosis, where elderly patients are known to have a higher probability of certain diseases. Spam filtering, where emails containing keywords like “prize” or “winner” are more likely to be spam. The ability to leverage such imperfect prior information stems from the robustness of postPU to inaccuracies in posterior estimates, as further elaborated in **A2**.
>
> We acknowledge that the current manuscript lacks sufficient detail on the construction of partitioning criteria and are augmenting it with more comprehensive experimental configurations with more meaningful criterions to clarify this aspect.
>
> **Q2: Weight Calculation and Dependence on Posterior Probabilities**
>
> **A2:** The reviewer rightly notes that the weights in postPU depend on posterior probabilities $\pi_i$ . We thank the reviewer for this observation and offer the following clarification:
>
> **Theoretical Robustness**: Although Eq. (5), (8) and (9) explicitly involves $\pi_i$, our method is designed to be robust to inaccuracies in these estimates as supported by **Theorem 3**. This theorem shows that even when the estimated posteriors $\hat{\pi}_i$ deviate substantially from the true $\pi_i$ , the risk estimator remains unbiased under the condition:
>
> $$\sum_{i=1}^{m}\lambda^+_i-\lambda^-_i=0$$
>
> Satisfying Eq. (5) is a **sufficient condition** for the above to hold. Therefore, the estimator performs well even with coarsely approximated $\hat{\pi}_i$ , as long as the weighting constraint is approximately met.
>
> **Empirical Validation**: In Section 6.2, we empirically validate this robustness. The results demonstrate that our risk estimator performs stably, and comparably to the oracle case, even when the provided posterior values deviate significantly from the true values. This confirms that the method is resistant to inaccuracies in class probability estimates.
>
> **Q3: Theorem Presentation and Notation Clarity**
>
> **A3:** We thank the reviewer for the suggestion to improve theorem presentation. In response, we now explicitly define essential notation from Table 7 prior to the respective theorems.

---

### Meta-Review · Area_Chair_T5eh · 2025-12-17

**Summary:**

This paper tackles an important limitation of prior-based PU learning and proposes a flexible posterior-based framework. However, key practical questions about how practitioners should construct subsets and obtain sufficiently informative posterior estimates remain under-specified, and the robustness experiments partially rely on oracle-style information. Despite scalability analysis and more baselines, the empirical validation is still on a narrow set of settings and does not convincingly establish advantages over existing PU methods. Therefore, I recommend reject.

BTW, I went through the paper and found that it has many mathematical/conceptual issues. I suggest being more serious about the formality of scientific writing.

For example, the authors used the mathcal version of R to denote 5 different concepts: the real number line, the true and empirical risks as functions of a model (a model is a function and then the risk is a functional), and the true and empirical risks as constants (i.e., risk functions evaluated at a specific model). In particular, after the authors "divide samples into m>=2 subsets", Eq. (2) says "following the idea of unbiased PU risk estimator defined by Eq.(1), we estimate PU risks in posterior-based PU learning setup by weighted sum of absolute risks in each subset", where Eq. (2) claims that the true risk over the test data distribution equals a weighted sum of the empirical risks over many subsets of training data samples, which is simply impossible. Even though Eq. (2) used R as constants, the meaning is still that R are functions of the model g(x) --- no matter g(x) is parametric or non-parametric, its function class is uncountable on cardinality; on the other hand, the freedom of the RHS of Eq. (2) is only 2m, which means that there are 2m variables but uncountably infinite equations. As a result, Eq. (2) is mathematically incorrect as the most important equation in the paper.

Moreover, class-prior-free methods are methods that don't make use of the class-prior probability p(y) of the test distribution, but it doesn't mean that the class-prior probability doesn't exist. PU learning just means PU classification and it is binary classification, where p(y) is not extremely small or large, such as between 0.1 and 0.9. If p(y) is smaller than 0.01 or larger than 0.99, it is no longer a binary classification problem; instead, it is a detection or retrieval problem. I don't say that the classification accuracy and error are the only proper performance measures in PU classification, but at least F1 and AUC are NOT proper performance measures since they are designed for detection and retrieval problems. Suppose, for the sake of argument, what the authors really care about is F1 or AUC because the problem of interest is not regarded as binary classification. Then, the authors should not take Eq. (1) as the starting point: unbiased risk estimators can only estimate the classification risk that is a surrogate performance measure of the classification error; if F1 or AUC matters more, the starting point should be unbiased F1 or AUC estimators, whose theoretical properties and empirical behaviors are completely different from unbiased risk estimators. Please be minded that two different real-world problems can share the same abstract mathematical formulation, while the specific value of some key parameter (the class prior p(y) in this case) determines to which problem a realization belongs.

**Reviewer Concerns:**

**q5ga** Concerns: practical procedure for data partitioning; how to obtain subset posteriors and weights; unclear theorem notation. Addressed: heuristic partition criteria with examples; robustness of the weighted risk to inaccurate posteriors. Outstanding: still lacks concrete guidance on constructing V and estimating posteriors/weights in new domains; the practicality beyond the given cases remains ambiguous.

**hc3X** Concerns: scalability analysis and runtime/memory comparison; narrow experimental domains. Addressed: Complexity analysis, runtime tables, discussion of memory costs.
Outstanding: broader domains are not convincingly demonstrated.

**LJUZ** Concerns: reliance on oracle-like posterior information; missing experiments on large m, beta-sensitivity, and recent baselines. Addressed: scalability and beta-sensitivity studies; more baseline comparisons. Outstanding: how practitioners obtain useful subset posteriors and V remains under-specified; comparisons to the full range of modern PU methods are limited.

**Lyij** Concerns: relation to class-prior-free PU methods; use of F1 instead of AUC; citation style. Addressed: conceptual comparisons; AUC results; citation formatting.
Outstanding: empirical comparisons to other class-prior-free methods beyond HolisticPU is still absent.

**Reviewer Scores:**

I don't think any of the four reviewers would like to (further) increase his or her score.

---

### Decision · Program_Chairs · 2026-01-26

Reject